# Influence of Carbon Content in Ni-Doped Mo₂C Catalysts on CO Hydrogenation to Mixed Alcohol

**Zhenjiong Hao [1], Xiaoshen Li [1], Ye Tian [1,*], Tong Ding [1], Guohui Yang [2], Qingxiang Ma [3], Noritatsu Tsubaki [2] and Xingang Li [1,4,*]**

[1]   Collaborative Innovation Center of Chemical Science and Engineering (Tianjin), State Key Laboratory of Chemical Engineering, Tianjin Key Laboratory of Applied Catalysis Science & Engineering, School of Chemical Engineering & Technology, Tianjin University, Tianjin 300350, China; 2017207306@tju.edu.cn (Z.H.); lixiaoshen@tju.edu.cn (X.L.); d_tong@tju.edu.cn (T.D.)

[2]   Department of Applied Chemistry, School of Engineering, University of Toyama, Gofuku 3190, Toyama 930-8555, Japan; thomas@eng.u-toyama.ac.jp (G.Y.); tsubaki@eng.u-toyama.ac.jp (N.T.)

[3]   State Key Laboratory of High-Efficiency Coal Utilization and Green Chemical Engineering, Ningxia University, Yinchuan 750021, China; maqx@nxu.edu.cn

[4]   School of Chemical and Biological Engineering, Lanzhou Jiaotong University, Lanzhou 730070, China

*   Correspondence: tianye@tju.edu.cn (Y.T.); xingang_li@tju.edu.cn (X.L.)

**Abstract:** Herein, we synthesize the Ni-doped Mo₂C catalysts by a one-pot preparation method to illuminate the effect of the number of carbon atoms in Mo₂C lattice on CO hydrogenation to mixed alcohol. The Ni doping inhibits the agglomeration of Mo₂C crystals into large particles and the surface carbon deposition, which increase the active surface area. In addition, the interaction between Ni and Mo increases the electron cloud density of Mo species and promotes the non-dissociative adsorption and insertion of CO. Especially, our results indicate that with the increase of the nickel content, the number of carbon atoms in Mo₂C lattice on the surface of the catalyst shows a volcano type variation. The low carbon content induces the formation of coordination unsaturated molybdenum species which exhibit the higher catalytic activity and mixed alcohol selectivity than other molybdenum species. Among the catalysts, the MC-Ni-1.5 catalyst with Ni/Mo molar ratio of 1.5:8.5, which has the largest amount of coordination unsaturated molybdenum species, shows the highest space-time yield of mixed alcohols, which is three times higher than that of the Mo₂C catalyst.

**Keywords:** Mo₂C; mixed alcohol synthesis; Ni doping; electron interaction; lattice carbon atoms

## 1. Introduction

Mixed alcohol can be used not only as clean fuels to replace fossil energy but also as solvents, intermediates of chemical raw materials and gasoline additives [1–4]. Considering energy utilization and environmental protection, directly producing mixed alcohol from syngas can realize clean and efficient utilization of coal resources [5,6].

Four kinds of catalysts are commonly used in CO hydrogenation to mixed alcohol: Rh-based catalysts [7–10], Mo-based catalysts [11–14], modified Fischer–Tropsch synthesis catalysts [15–18] and modified methanol synthesis catalysts [19,20]. Molybdenum carbide catalysts are widely used in CO or CO₂ hydrogenation to mixed alcohol [21–23], water gas shift [24,25], methanol steam reforming [26–28], methane dry reforming [29] and so on. At present, the research results show that molybdenum carbide catalysts have excellent carbon deposition resistance and sulfur resistance [30]. However, CO hydrogenation to mixed alcohol is often accompanied by side reactions such as Fischer–Tropsch synthesis to hydrocarbon and water gas shift. Therefore, it is necessary to improve the selectivity of alcohol by catalyst modification [31–36].

Some researchers found that the performance of the molybdenum carbide catalysts for CO hydrogenation to mixed alcohol could be improved by transition metal modification [37–39]. Two main reasons are considered to improve the performance of the catalyst.

The new formed mixed carbide of metal and Mo, such as $Ni_6Mo_6C$ [40] and $Co_3Mo_3C$ [41], might be active sites for alcohol synthesis. The other reason is that the metal–support interaction balances dissociative adsorption and molecular adsorption of CO [42] and further promotes formation of alcohol. However, due to the complexity of the reaction, there is not enough evidence to confirm these views, and the reason for improvement of catalyst performance is still unclear. To identify the factors affecting the catalyst performance for CO hydrogenation to mixed alcohol is still a great challenge.

Defects often play an important role in catalytic reactions [43–45]. Some researchers found that there are many molybdenum and carbon defects in molybdenum carbide [43,45–47], which might induce electronic and structural changes [48]. However, at present, the effect of these defects on the catalytic performance for CO hydrogenation is still unclear. Although these defects are still difficult to characterize, they are obviously related to the number of carbon atoms in the crystal lattice. Therefore, the effect of defects can be studied by studying carbon content in molybdenum carbide, which will be helpful for rational design of high-performance molybdenum carbide catalyst.

In this work, the Ni and Mo mixed oxide precursors were prepared by a one-pot method, and the Ni-doped molybdenum carbide catalysts were synthesized by the carbonization of the precursors. In order to avoid the formation of local hot spots on the catalyst, which will lower the selectivity of alcohol, we adopted slurry bed reactor to evaluate the catalytic performance of the catalysts for CO hydrogenation to mixed alcohol [49,50]. X-ray photoelectron spectroscopy (XPS), high resolution transmission electron microscopy (HRTEM), scanning electron microscopy (SEM), X-ray powder diffraction (XRD), $N_2$ physical adsorption and thermogravimetry (TG) were used to investigate the structure–activity relationship of the catalyst in the reaction system. Especially, the effect of the number of carbon atoms in molybdenum carbide lattice on the catalytic performance was discussed.

## 2. Results and Discussion

### 2.1. Physical Properties of the Catalysts

Figure 1 shows the XRD patterns of the catalysts. The diffraction peaks at $2\theta = 34.4$, 38.0, 39.5, 52.3, 61.7, 69.7 and 74.9° belong to β-$Mo_2C$ phase (PDF#72-1683) [28]. For the MC catalyst, only the diffraction peaks of β-$Mo_2C$ phase were detected. For the MC-Ni-X catalysts, besides the β-$Mo_2C$ phase, a new diffraction peak appeared at $2\theta = 44.4°$. It can be attributed to n-diamond type carbon phase (PDF#43-1104). No diffraction peak of Ni species is observed in Figure 1, indicating that Ni species are highly dispersed in the catalysts. The enlarged XRD patterns (Figure 1B) show that with the increase of the nickel content, the diffraction peak at $2\theta$ of 39.5° gradually shifts to higher angles. It indicates that $Ni^{2+}$ cations with smaller ionic radius than $Mo^{2+}$ cations are doped into the molybdenum carbide lattice, which makes the crystal plane spacing smaller [51]. Compared with the MC catalyst, the diffraction peak intensity of the β-$Mo_2C$ phase of the MC-Ni-X catalysts increases sharply. It suggests that the crystallinity of β-$Mo_2C$ crystals in the Ni-doped catalysts significantly increases. The average size of β-$Mo_2C$ crystals was calculated by Scherrer formula, as listed in Table 1. The results show that the size of β-$Mo_2C$ crystal in the MC-Ni-X catalysts is about twice larger than that of the MC catalyst.

**Table 1.** Physical properties of the catalysts.

| Catalysts | [a] Ni:Mo | [b] $d$ (nm) | [c] $S_{BET}$ ($m^2 g^{-1}$) | [d] C/Mo |
|:---:|:---:|:---:|:---:|:---:|
| MC | - | 10.2 | 17.5 | 1.4 |
| MC-Ni-1 | 1:9 | 19.2 | 2.6 | 0.75 |
| MC-Ni-1.5 | 1.5:8.5 | 18.7 | 3.4 | 0.85 |
| MC-Ni-2 | 2:8 | 24.7 | 2.2 | 0.96 |

[a] Molar ratio of Ni and Mo. [b.] The size of $Mo_2C$ crystallite, calculated by Scherrer formula according to the peak at $2\theta = 39.5°$ in the XRD patterns. [c] BET surface area obtained by nitrogen adsorption method. [d] Determined by TG.

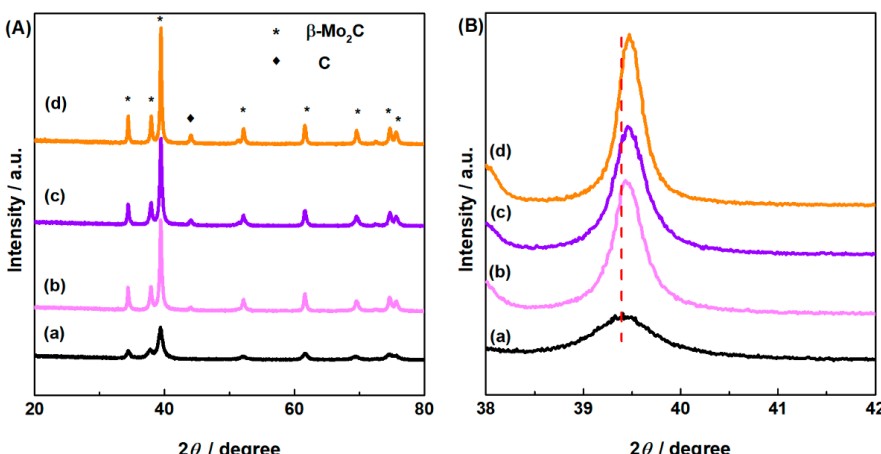

**Figure 1.** XRD patterns of (a) MC, (b) MC-Ni-1, (c) MC-Ni-1.5 and (d) MC-Ni-2 in the 2θ range of (**A**) 20–80° and (**B**) 38–42°.

Figure 2 shows the HRTEM images of the catalysts. We observed the (121), (200) and (021) planes of $Mo_2C$ in MC catalyst, but only observed (121) plane in Ni-doping catalysts. Figure 2a shows that some carbon is deposited on the surface of the MC catalyst, and the amount of carbon deposition on the surface of the MC-Ni-X catalysts is significantly decreased by Ni doping. No Ni-related crystal is observed in the TEM images of the MC-Ni-X catalysts, indicating that Ni atoms are doped in $Mo_2C$, which is consistent with the XRD results.

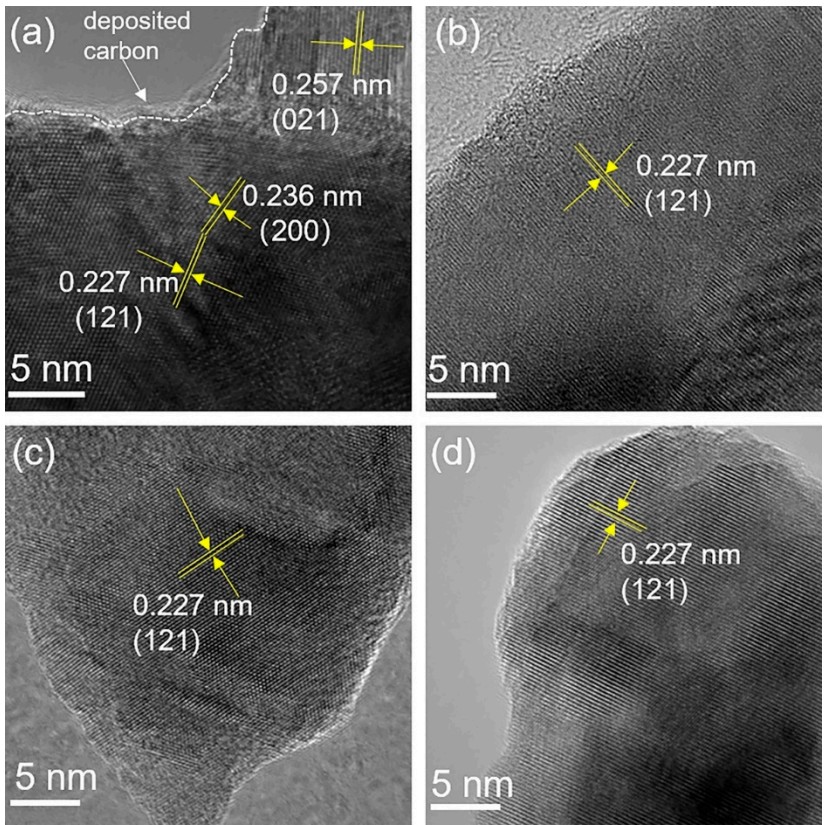

**Figure 2.** HRTEM images of (**a**) MC, (**b**) MC-Ni-1, (**c**) MC-Ni-1.5 and (**d**) MC-Ni-2.

Figure 3 shows the SEM images of the catalysts. The particles of the MC catalyst in Figure 3a present the irregular large lamellar structure, and most of them are about 1 μm in diameter and 100–200 nm in thickness. After the Ni doping, the particle size of the catalyst

in Figure 3b–d significantly decreases, and most of the particles are less than 100 nm. Many stacking pore structures are formed in the MC-Ni-X catalysts. However, the XRD and TEM results show that the crystal size of the MC catalyst is smaller than that of the MC-Ni-X catalysts. Combining the results of XRD and SEM, we believe that small crystals in the MC catalyst finally agglomerate into large particles. Nevertheless, the Ni doping significantly inhibits the agglomeration of the $Mo_2C$ crystal [40] which is conducive to increasing the active surface area and further promoting the catalytic performance.

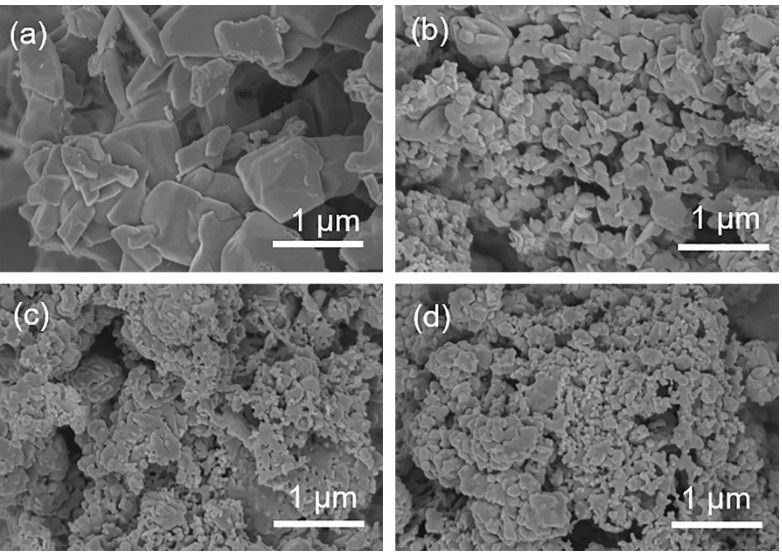

**Figure 3.** SEM images of (**a**) MC, (**b**) MC-Ni-1, (**c**) MC-Ni-1.5 and (**d**) MC-Ni-2.

Table 1 gives the specific surface area calculated from the nitrogen adsorption-desorption isotherms. The specific surface area of the MC-Ni-X catalysts is about 3.0 $m^2$ $g^{-1}$, which is only one sixth of that of the MC catalyst (17.5 $m^2$ $g^{-1}$). The isotherm of the MC catalyst (Figure S1 in Supplementary Materials) is similar to that of amorphous carbon [52,53]. We infer that the larger specific surface area of the MC catalyst is due to more carbon deposition on it than that on the MC-Ni-X catalysts.

## 2.2. Catalytic Performances

Table 2 provides the activity of the catalysts and the selectivity of mixed alcohol. The selectivity of mixed alcohol over the MC-Ni-X catalysts is significantly higher than that over the MC catalyst. It demonstrates that the Ni doping of $Mo_2C$ can promote the formation of mixed alcohol products. For the MC-Ni-X catalysts, with the increase of the Ni content, the CO conversion decreases, while the selectivity of mixed alcohol increases. The space-time yield of mixed alcohol over the MC-Ni-1.5 catalyst is the highest among the catalysts. Moreover, at the similar catalytic activity, the reaction temperature of the MC-Ni-1.5 catalyst decreases by more than 70 °C compared with the reported molybdenum carbide-based catalysts [8,37,42,54] (Table 3). It is probably because the reaction environment of the slurry bed reactor is significantly different from that of the fixed bed one [49]. We also calculated the $r_s$ (CO conversion rate per unit specific surface area) of each catalyst, and the results are given in Table 2. It can be seen from the table that the $r_s$ of the MC catalyst is much smaller than that of the Ni-doped catalysts.

**Table 2.** The catalytic performance of the catalysts.

| Catalysts | [a] $X_{CO}$ (%) | [b] $r_s$ (mmol·$g_{cat}^{-1}$·$h^{-1}$·$m^{-2}$·g) | [c] $S$ (%) | | [d] HC | ROH Distribution (%) | | [e] $STY_{ROH}$ (mg $g_{cat}^{-1}$ $h^{-1}$) |
|---|---|---|---|---|---|---|---|---|
| | | | **MeOH** | **$C_{2+}$OH** | | **MeOH** | **$C_{2+}$OH** | |
| MC | 4.4 | 0.067 | 10.7 | 31.1 | 58.2 | 25.6 | 74.4 | 7.5 |
| MC-Ni-1 | 9.7 | 1.179 | 29.1 | 30.7 | 40.2 | 48.7 | 51.3 | 24.3 |
| MC-Ni-1.5 | 7.4 | 0.707 | 35.8 | 26.2 | 38.0 | 57.7 | 42.3 | 25.0 |
| MC-Ni-2 | 4.4 | 0.634 | 38.5 | 29.8 | 31.7 | 56.3 | 43.7 | 14.4 |

[a] CO conversion. [b] $r_s$ refers to CO conversion rate per unit specific surface area. [c] Selectivity: free of $CO_2$. [d] Hydrocarbons. [e] Space-time yield of mixed alcohol. Reaction conditions: $T$ = 180 °C, $P$ = 5 MPa, $H_2$:CO = 2:1, 0.69 g catalysts, $V$=20 mL $min^{-1}$, using dioxane (50 mL) as solvent.

**Table 3.** Comparison with the activity of the catalysts in references for MAS.

| Catalysts | [a] $X_{CO}$ (%) | [b] $S$ (%) | | | [c] $STY_{ROH}$ (mg·$g_{cat}^{-1}$·$h^{-1}$) | [d] $T$, $P$ | $H_2$/CO | Reactor Type | Ref. |
|---|---|---|---|---|---|---|---|---|---|
| | | **MeOH** | **$C_{2+}$OH** | **HC** | | | | | |
| 6.1%Ni/K/$Mo_2$C | 3.7 | 7.8 | 34.8 | 57.4 | 24.5 | 250, 4.0 | 2 | fixed bed | [42] |
| 6.5%Cu/K/$Mo_2$C | 7.3 | 6.7 | 24.2 | 69.1 | 35.4 | 280, 4.0 | 2 | fixed bed | [42] |
| 1.5%Rb/$Mo_2$C/$Al_2O_3$ | 5.0 | 16.0 | 40.0 | 44.0 | - | 300, 3.0 | 1 | fixed bed | [37] |
| $K_2CO_3$/$Mo_2$C | 3.8 | 20.1 | 30.4 | 49.5 | 20.0 | 250, 10 | 1 | fixed bed | [54] |
| 1%Rh/$Mo_2$C/$SiO_2$ | 1.5 | 60.0 | 20.0 | 20.0 | - | 250, 5.8 | 1 | fixed bed | [8] |
| MC-Ni-1.5 | 7.4 | 35.8 | 26.2 | 38.0 | 25.0 | 180, 5.0 | 2 | slurry bed | This work |

[a] CO conversion. [b] Selectivity: free of $CO_2$. [c] Space-time yield of mixed alcohol. [d] Reaction temperature (°C) and pressure (MPa).

## 2.3. Surface Properties of the Catalysts

Figure 4 shows the Mo 3*d* XPS spectra of the catalyst. There are four binding energy peaks in the Mo $3d_{5/2}$ region of the MC catalyst. The peaks at 228.5, 229.1, 231.3 and 232.8 eV are assigned to $Mo^{2+}$, $Mo^{4+}$, $Mo^{5+}$ and $Mo^{6+}$, respectively [55,56]. The MC-Ni-X catalysts have a new binding energy peak at 227.0 eV, which belongs to $Mo^0$. Table 4 lists the relative contents of various valence Mo species on the catalyst surface calculated according to the XPS peak area. Compared with the MC catalyst, the peak position of $Mo^{2+}$ on the MC-Ni-X catalysts shifted about 0.2 eV towards lower binding energy. It indicates that there is partial electron transfer from Ni to Mo, which increases the electron cloud density of Mo species in the catalyst. It is beneficial to non-dissociative adsorption and insertion of CO to produce mixed alcohol [40,42].

It has been proved that the molybdenum species in the $Mo_2$C bulk are $Mo^{2+}$ [56]. In addition, there exists some low valence molybdenum species ($Mo^{\delta+}$, $0 < \delta < 2$) on the $Mo_2$C surface under synthesis and reaction conditions because the $Mo_2$C catalyst is synthesized in a strong reduction atmosphere of methane and hydrogen at 700 °C. After passivation, the surface of the catalyst was partially oxidized, so the higher valence molybdenum species appear in the ex-situ XPS results, including $Mo^{4+}$, $Mo^{5+}$ and $Mo^{6+}$.

Figure 5 shows the schematic diagram of the surface states of the catalysts. The first case is the formation of $Mo^0$ on the surface due to the excessive reduction in the process of carbonization. The second and third case are the formation of the carbon defects around the molybdenum atoms and the molybdenum defects around the carbon atoms, respectively.

**Table 4.** The percentage content of Mo species calculated from the XPS spectra.

| Catalysts | **Mo 3*d*** | | | | |
|---|---|---|---|---|---|
| | **$Mo^0$** | **$Mo^{2+}$** | **$Mo^{4+}$** | **$Mo^{5+}$** | **$Mo^{6+}$** |
| MC | 0 | 42.7 | 21.7 | 8.8 | 26.8 |
| MC-Ni-1 | 13.4 | 43.6 | 23.9 | 10.9 | 8.2 |
| MC-Ni-1.5 | 4.6 | 50.2 | 28.3 | 11.9 | 5.0 |
| MC-Ni-2 | 2.8 | 55.6 | 23.3 | 9.0 | 9.3 |

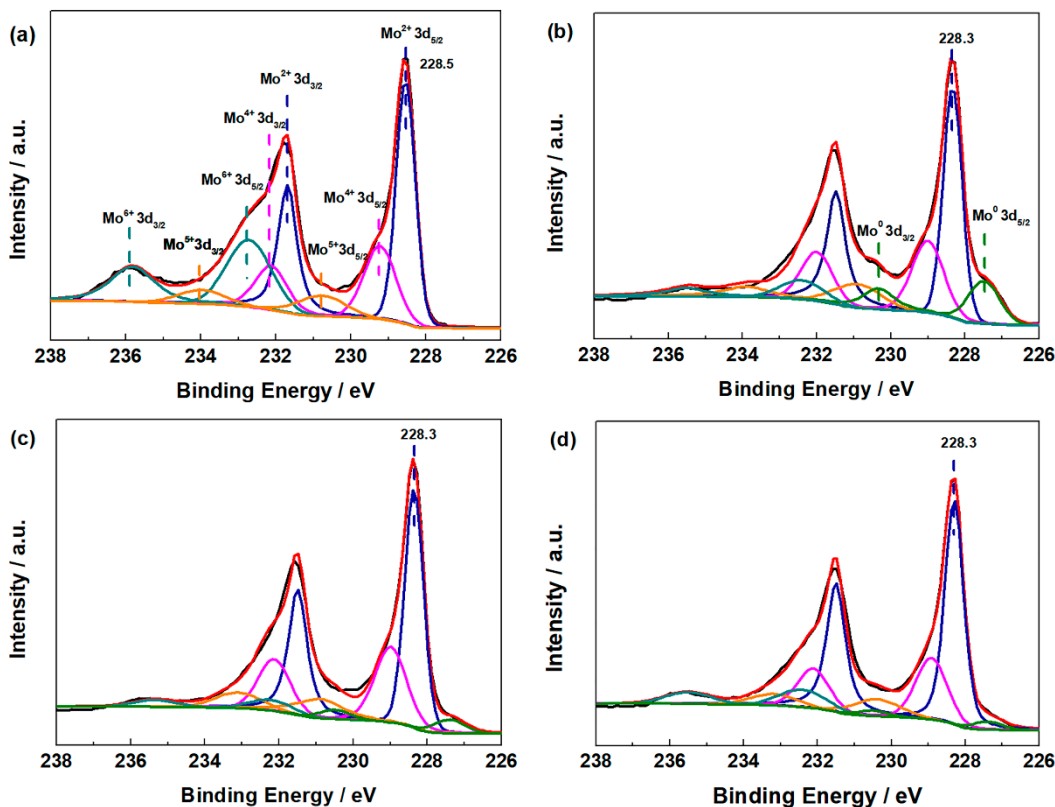

**Figure 4.** Mo $3d$ XPS spectra of (**a**) MC, (**b**) MC-Ni-1, (**c**) MC-Ni-1.5 and (**d**) MC-Ni-2. $Mo^0$: green line, $Mo^{2+}$: blue line, $Mo^{4+}$: pink line, $Mo^{5+}$: orange line, $Mo^{6+}$: cyan line.

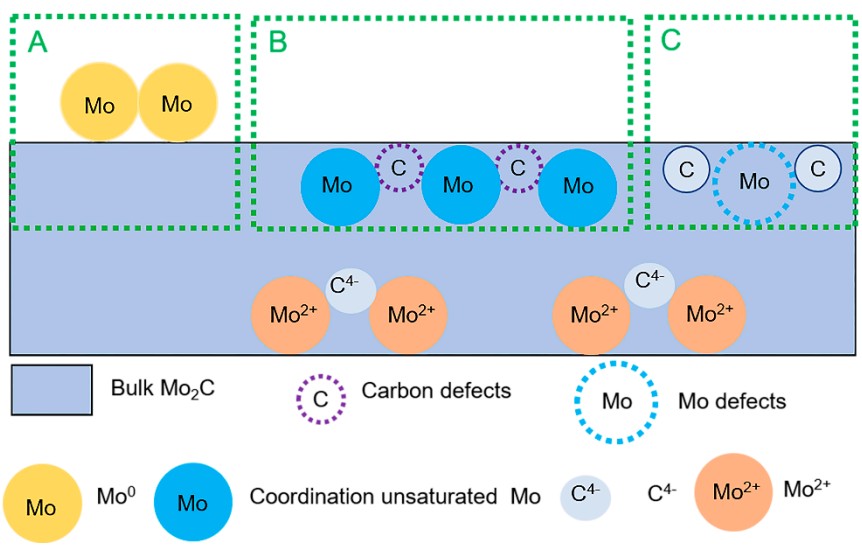

**Figure 5.** Schematic diagram of $Mo_2C$ surface. (**A**) $Mo^0$, (**B**) carbon defects and (**C**) Mo defects.

The surface $Mo^0$ species are more easily oxidized than other Mo species in Figure 5. Thus, we assigned $Mo^{6+}$ species in Figure 4 to the oxidation of $Mo^0$ during passivation. The XPS results show that the $Mo^{6+}$ species on the surface of the MC-Ni-X catalysts is only 1/3 to 1/5 to that of the MC catalyst. The activity results suggest that the $Mo^0$ species is not the active site of the catalyst, since the MC-Ni-X catalysts have higher activity than the MC catalyst.

Similarly, Mo species near carbon defects is readily oxidized than stoichiometric $Mo_2C$. Thus, we assume that $Mo^{4+}$ and $Mo^{5+}$ species in Figure 4 are originated from Mo species

around carbon defects. The activity results show the catalyst with higher $Mo^{4+}$ and $Mo^{5+}$ content of has higher space-time yield of mixed alcohol. It indicates that these Mo species have higher activity for the synthesis of mixed alcohol than other Mo species.

### 2.4. TG Analysis

Figure 6 gives the TG and derivative thermogravimetry (DTG) curves of the catalysts. The DTG curve of the MC catalyst can be divided into five temperature ranges, which are marked as stage A, B, C, D and E. In order to determine the oxidation process corresponding to each temperature range, the samples obtained by oxidizing the MC catalyst at different temperatures (MC-T) were characterized by XRD.

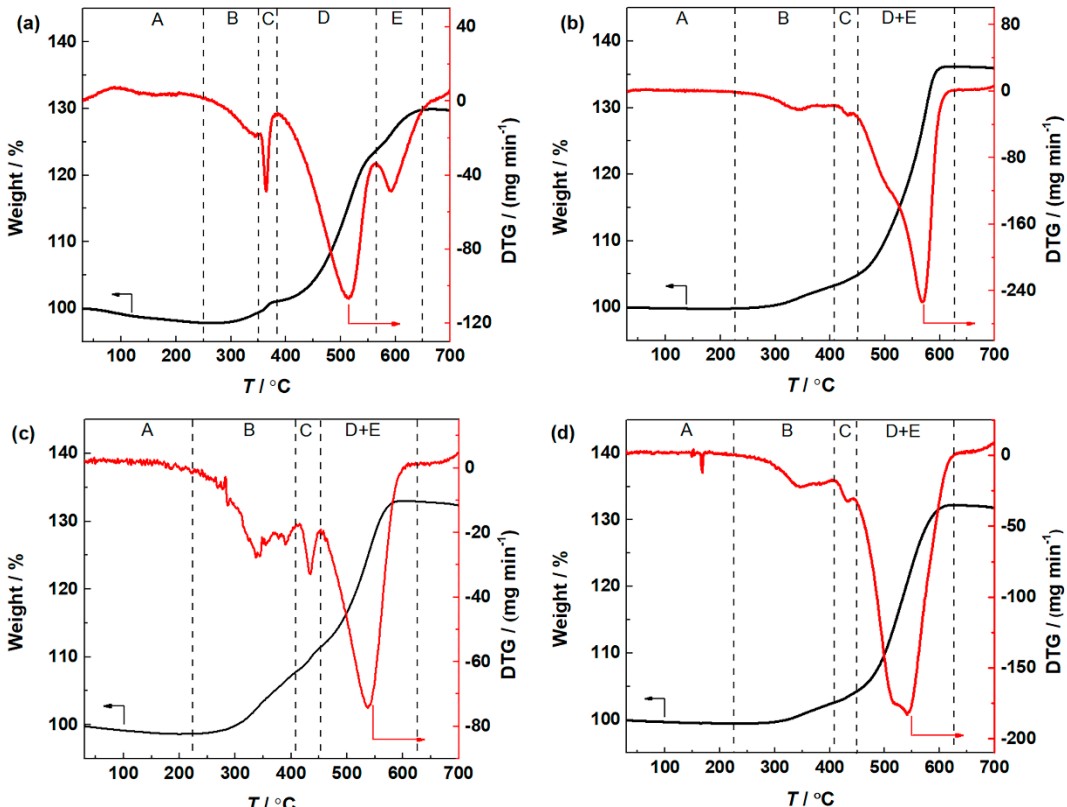

**Figure 6.** TG curves of (**a**) MC, (**b**) MC-Ni-1, (**c**) MC-Ni-1.5 and (**d**) MC-Ni-2.

Figure 7 shows the XRD patterns of these samples. The selected oxidation temperature points were determined according to the DTG curve (Figure S2 in Supplementary Materials) of the MC catalyst with a slow heating rate (1 °C min$^{-1}$) to ensure that the selected temperature points are more illustrative. The MC-270 and MC-305 samples were further characterized by XPS to determine the surface species.

The weight loss below 250 °C can be attributed to the removal of adsorbed impurities on the catalyst surface. The MC-270 sample corresponds to stage B of the DTG curve. According to the XRD results (Figure 7), the composition of the MC-270 sample is similar to the MC catalyst, which exists as the β-$Mo_2C$ phase (PDF#72-1683). However, the XPS results (Figure S3 in Supplementary Materials) show that the surface of the MC-270 sample is mainly composed of $Mo^{6+}$ and $Mo^{5+}$, indicating that the catalyst surface has been oxidized to molybdenum oxide. Therefore, we attribute the weight gain in stage B to the oxidation of surface $Mo_2C$. Since the oxidation temperature of stage B is relatively low, the carbon deposited on the surface has not been oxidized, so it can reflect the number of carbon atoms in the surface molybdenum carbide lattice.

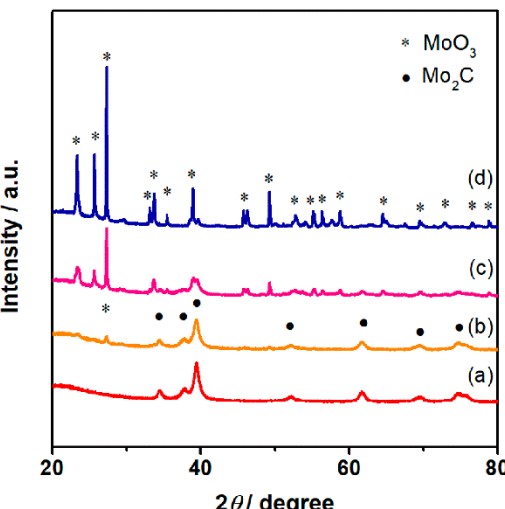

**Figure 7.** XRD patterns of (a) MC-270, (b) MC-305, (c) MC-450 and (d) MC-550.

The MC-305 sample corresponds to stage C of the DTG curve. According to the XRD results (Figure 7), the MC-305 sample still mainly exists as $Mo_2C$. At the same time, a very weak peak belonging to $MoO_3$ phase (PDF#01-0706) appeared at $2\theta$ of 27.2°. We attribute it to the oxidation of small $Mo_2C$ particles highly dispersed on the surface of large catalyst particles. The MC-450 and MC-550 sample correspond to stage D and E of the DTG curve, respectively. The XRD patterns of the two samples agree well with the $MoO_3$ phase (PDF#01-0706). The XRD peak intensity of the MC-550 sample is significantly stronger than that of the MC-450 sample. The weight gain at the two stages can be attributed to the oxidation of the bulk $Mo_2C$ to $MoO_3$. For these two stages, the product is the same, but the DTG curve shows two peaks. We speculate that the rate control step of the oxidation process has changed. Stage D is the oxidation of the outer $Mo_2C$. Stage E is the oxidation of the internal $Mo_2C$, and in this stage, the rate control step of oxidation is that oxygen enters from outside to inside of crystals.

The DTG curves of the MC-Ni-X catalysts are similar to that of the MC catalyst, but the boundary between the stage D and E is not obvious, probably due to the smaller particle size of the former catalysts than that of the latter one.

According to the TG curves, the amount of weight gain at different stages was calculated. The weight gain of stage B corresponds to the number of carbon atoms in the surface molybdenum carbide lattice of the catalyst, and the total weight gain of stage C, D and E corresponds to the bulk carbon content of the catalyst. The less weight gain indicates the more carbon content. Figure 8 shows the surface, bulk phase and total weight gain of the catalysts. With the increase of the nickel content, the weight gain shows a volcano type variation, indicating that the surface carbon content obeys the similar tendency, which is consistent with the trend of surface $Mo^{4+}$ and $Mo^{5+}$ contents and the space-time yield of mixed alcohol. This demonstrates that the number of carbon atoms in the surface molybdenum carbide lattice may be the key factors affecting the synthesis of mixed alcohol.

Table 1 provides the molar ratio of C and Mo (C/Mo) in the catalyst calculated from the TG results. The C/Mo ratios of all the catalysts exceed 0.5, which is the C/Mo theoretical value of $Mo_2C$. It may be due to the fact that we assume the weight decrease of stage A only to desorption during TG analysis. However, a small amount of surface molybdenum species may be oxidized therein, which makes us overestimate the carbon content in the catalyst. On the other hand, the higher C/Mo ratio than 0.5 can also be attributed to carbon deposition on the catalyst surface.

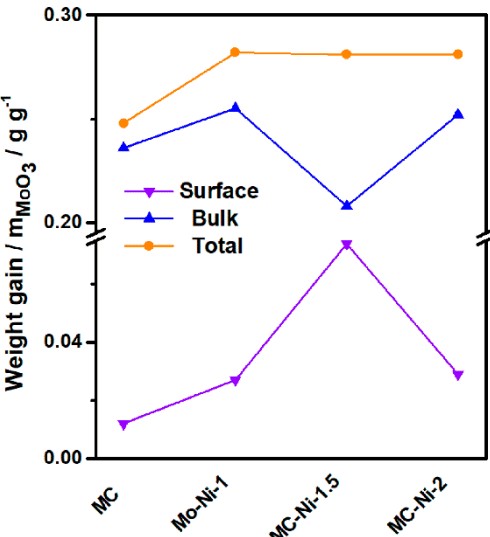

**Figure 8.** The ratios of the surface, bulk phase and total weight gain of the catalysts to the total weight of $MoO_3$ obtained by the complete oxidation of the catalysts, calculated from the TG analysis results.

The C/Mo ratio of the MC catalyst is 1.4 times higher than that of the MC-Ni-X catalyst. The Ni doping significantly decreases the amount of carbon on the surface and in the bulk of the catalyst during the carbonization of the precursor. It can accordingly reduce the numbers of carbon atoms in the molybdenum carbide lattice and active sites exposed on the catalyst surface. This inevitably promotes the activity of the Ni-doped catalysts for CO hydrogenation.

## 3. Materials and Methods

### 3.1. Catalysts Preparations

Typically, the catalyst precursors were prepared by a one-pot method. In a mixed solvent of deionized water and ethanol, 0.1 mol of citric acid ($C_6H_8O_7$) was dissolved. To the citric acid solution was added 0.1 mol (the total amount of Ni and Mo atoms) of $Ni(NO_3)_2 \cdot 6H_2O$ and $(NH_4)_3Mo_7O_{24} \cdot 4H_2O$ according to the molar ratio of Ni:Mo = X:(10-X), (X = 1, 1.5, 2). The suspension was stirred until $Ni(NO_3)_2$ and $(NH_4)_3Mo_7O_{24}$ were completely dissolved with addition of ammonia to get a stable pH value of 3 and then evaporated in oil bath at 120 °C for 4 h. Then, it was dried overnight in a constant temperature oven at 100 °C for 12 h. The precursor was obtained by calcining the solid powder in a muffle furnace at 500 °C for 3 h.

The precursors were carbonized to obtain the Ni-doped molybdenum carbide catalysts in a tube furnace in 20% $CH_4/H_2$ flow (100 mL min$^{-1}$). Precursor (0.69 g) was heated from room temperature to 700 °C and maintained for 2 h. The obtained catalyst was cooled to 40 °C naturally. For comparison, molybdenum carbide catalyst was also prepared. The preparation process was consistent with the above process except the absence of nickel. For convenience, the Ni-doped molybdenum carbide catalyst was named as MC-Ni-X, (X = 1, 1.5, 2), and the prepared $Mo_2C$ catalyst was named as MC.

In order to investigate the carbonization of the precursor, several samples were prepared by oxidizing the MC catalyst at different temperatures. MC catalyst (80 mg) was heated to the specified temperature of $T$ in air flow (100 mL min$^{-1}$) at a heating rate of 1 °C min$^{-1}$, and kept at temperature $T$ for 20 min. The resulting sample was named as MC-T.

### 3.2. Catalysts Characterization

Before the catalyst characterization, all catalysts were added to dioxane for passivation under the protection of $CH_4/H_2$.

The XRD test was carried out on a D8-focus X-ray diffractometer (Bruker, Karlsruhe, Germany) with the emission source of Cu $K\alpha$ ($\lambda$ = 0.15418 nm). The nitrogen physical adsorption test was carried out on a QuadraSorb SI physical adsorption instrument (Quantachrome Instruments, Boynton Beach, FL, USA). The SEM images of the catalysts were taken on a Hitachi S-4800 scanning electron microscope (Hitachi S-4800, Tokyo, Japan). The high-resolution transmission electron microscope (HRTEM) images of the catalyst were obtained on a JEOL-JEM-2100F electron microscope (JEOL, Tokyo, Japan). The XPS experiments were conducted on A K-Alpha+ X-ray photoelectron spectrometer (Thermo Fisher Scientific, Walham, MA, USA). Al $K\alpha$ was used as X-ray source.

Thermogravimetric analysis was performed on a STA7300 thermogravimetric analyzer. The catalyst was heated to 700 °C at the heating rate of 10 °C min$^{-1}$ in air flow (100 mL min$^{-1}$). Based on the thermogravimetric results, the molar ratio of C to Mo (C/Mo) was calculated by the following equation:

$$x = m_2/[M_{Mo} + pM_{Ni} + (p + 3)M_O] \tag{1}$$

$$y = px \tag{2}$$

$$z = (m_1 - xM_{Mo} - yM_{Ni})/M_C \tag{3}$$

$$q = z/y \tag{4}$$

where $x$, $y$ and $z$ are the molar quantities of Mo, Ni and C, respectively. $p$ and $q$ are Ni/Mo and C/Mo molar ratio, respectively. $m_1$ is the mass of the catalyst after removing the adsorbed impurities at the temperature of 200–250 °C. $m_2$ is the total mass of $MoO_3$ and NiO obtained by complete oxidation of Mo and Ni species in the catalyst at 600–650 °C. $M_{Mo}$, $M_{Ni}$, $M_O$ and $M_C$ are the relative atomic mass of Mo, Ni, O and C, respectively.

The ratios of the surface, bulk phase and total weight gain of the catalysts to the total weight of $MoO_3$ ($m_{MoO3}$) obtained by the complete oxidation of the catalysts were also calculated from the TG analysis results.

### 3.3. Activity Measurement

CO hydrogenation experiments were carried out in a micro slurry bed reactor. Dioxane was used as the reaction solvent and 2-butanol was used to absorb the liquid products in a cold trap. Catalysts (0.69 g) was added to the slurry bed reactor in the protection of synthesis atmosphere. The gas flow rate was 20 mL min$^{-1}$. The gas composition was CO (30%), $CO_2$ (5%), $H_2$ (62%) and Ar (3%). The reaction pressure was 5 MPa, and the reaction temperature was 180 °C. The gas chromatograph (GC-9060) produced by Shanghai Ruimin company was used for on-line detection of tail gas. The organic components were detected by a flame ionization detector (FID), and inorganic components were detected by a thermal conductivity detector (TCD). The liquid products were quantitatively detected by a gas chromatography–mass spectrometry (GCMS-QP2010) produced by Shimadzu Company (Kyoto, Japan).

The conversion, selectivity and space-time yield of mixed alcohol ($STY_{ROH}$) were calculated by the following equations:

$$X_{CO} = (n_{in} - n_{out})/n_{in} \times 100\% \tag{5}$$

$$S_i = m_i \times n_i/[\sum(m_{HC} \times n_{HC}) + \sum(m_{ROH} \times n_{ROH})] \times 100\% \tag{6}$$

$$STY_{ROH} = M_{ROH}/(T_{rec} \times m_{cat}) \tag{7}$$

where $X_{CO}$ is the conversion of CO, $n_{in}$ and $n_{out}$ are the molar flow rates of CO in the inlet and outlet gas, respectively. $S_i$ is the selectivity of product i, and $m_i$ and $n_i$ are the number of carbon atoms and moles of product i, respectively. $n_{HC}$ and $n_{ROH}$ are the mole numbers of hydrocarbons and alcohols in the product, respectively. $m_{HC}$ and $m_{ROH}$ are the corresponding carbon atom numbers of hydrocarbons and alcohols, respectively. $M_{ROH}$ is the mass of mixed alcohol, $T_{rec}$ is the reaction time, $m_{cat}$ is the mass of the catalyst, respectively.

## 4. Conclusions

In this work, the catalyst precursors were prepared by a one-pot method and then carbonized to obtain the Ni-doped $Mo_2C$ catalysts. With the protection of synthesis atmosphere, the catalyst was transferred to a slurry bed reactor to test the activity of CO hydrogenation to mixed alcohol. The Ni doping promotes not only the CO hydrogenation activity but also the selectivity of mixed alcohol by more than 20% compared with $Mo_2C$. The SEM, TEM and TG results show that the Ni doping inhibits the agglomeration of $Mo_2C$ crystals and the surface carbon deposition to increase the active surface area. The XPS and TG results show that the Ni doping decreases the number of carbon atoms in the molybdenum carbide lattice on the catalyst surface to simultaneously form coordination unsaturated surface Mo species, which is more active for CO hydrogenation. In addition, we discover the electron transfer from Ni to Mo, which increases the electron cloud density of Mo species, improving CO non-dissociation adsorption and insertion to produce mixed alcohol products.

**Supplementary Materials:** The following are available online at https://www.mdpi.com/2073-4344/11/2/230/s1. Figure S1: $N_2$ adsorption-desorption isotherms of the catalysts. (a) MC; (b). MC-Ni-1; (c) MC-Ni-1.5; (d) MC-Ni-2. Figure S2: TG curves of MC with the heating rate of 1 °C $min^{-1}$ in air. 270, 305, 450 and 550 °C are chosen to represent stage B, C, D and E, respectively. Figure S3: Mo 3*d* XPS spectra of (a) MC-270 and (b) MC-305.

**Author Contributions:** Z.H., investigation, writing—original draft; Y.T., investigation, writing—review and editing, supervision; X.L.(Xiaoshen Li), investigation, methodology; X.L.(Xingang Li), funding acquisition, writing—review and editing, project administration, resources, supervision; T.D., resources; G.Y., methodology; Q.M., methodology; N.T., resources, methodology. All authors have read and agreed to the published version of the manuscript.

**Funding:** This work was supported by National Natural Science Foundation of China (No. 21676182), the Program for Introducing Talents of Discipline to Universities of China (No. BP0618007), and State Key Laboratory of High-efficiency Utilization of Coal and Green Chemical Engineering (2020-KF-26).

**Data Availability Statement:** No new data were created or analyzed in this study.

**Acknowledgments:** The authors greatly acknowledge the support by the Large Instrument Center of the School of Chemical Engineering, Tianjin University.

**Conflicts of Interest:** The authors declare no conflict of interest.

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
