# Peer review of "Influence of Carbon Content in Ni-Doped Mo2C Catalysts on CO Hydrogenation to Mixed Alcohol"

_catalysts, doi:10.3390/catal11020230_

Round 1
Reviewer 1 Report
The current work describes synthesis of Ni doped Mo2C catalysts, which were characterized using XPS, TGA, XRD, SEM, and HRTEM. Subsequently, the catalysts were used in a slurry reactor bed to test their CO hydrogenation into mixed alcohol. Based on their results, the authors propose that the displacement of C atoms on the surface with Ni improves surface area by preventing large Mo2C crystal formation; furthermore, overall increased electron density results in increased activity and selectivity for mixed alcohols. The results supports the arguments and the data is presented clearly.
Reviewer 2 Report
This manuscript reported how the carbon defects affected the CO hydrogenation to mixed alcohol, and Ni addition could promote alcohol selectivity. The study pointed out some evidence that carbon deposition reduction and the suitable Mo oxidation state could enhance the amount of alcohol. However, there remain some matters that need to be elucidated. The manuscript can be accepted in Catalysts after some revisions.
- In the final paragraph of page 3, the grammar error is noted in “Figure 2 and show…” sentence.
- In Table 1, the calculation of Mo2C crystallite based on 2θ=39.4o should be corrected by 2θ=39.5o as XRD patterns description.
- In figure 2b-c, how is about the spacing of other fringes in different directions? If there is only (121) plane observation, the author should note and describe it in the manuscript.
- In part 2.2 on page 5, the author mentioned the reaction temperature difference observed due to the kinds of reactors. A comparative table should be made here to evaluate MC-Ni-X samples of this study with the previous MC-Ni samples.
- How is the catalyst regeneration ability? The Mo4+ and Mo5+ seemed to promote the high selectivity of alcohol products, so XPS analysis of used samples could be carried out to claim the Mo oxidation state changes.
Reviewer 3 Report
The authors have examined a series of Ni-containing MoC’s for alcohol synthesis. Frankly, I think more work is needed before this should be published. There is far too little description of the reactions and no attempt at benchmarking this to what is in the literature.
Specific comments are shown below:
1) The English is not terrible but there are many places where it is stilted.
2) Table 1 should include compositions. What is MC-Ni-1.5? It is not good to ask the reader to go to the back to find out what is being described. Frankly, Vp is not particularly meaningful with catalysts that have surface areas of 2 m2/g. Furthermore, while I understand that the journal would like the experimental methods section to be at the end, there needs to be some description of how the samples were pretreated prior to performing the characterization methods. For example, even conventional Ni catalysts with high Ni loadings usually do not show an XRD peak for Ni unless the catalyst has been reduced.
3) For materials with surface areas this low, it must be recognized that, since the bulk is not involved in the reaction, the characterization methods that focus on bulk properties are of questionable use. I am not saying these should not be done but there needs to be more recognition that the information may not be all that useful. Furthermore, it does not make sense to talk about diffusion limitations at the particle level for materials with surface areas this low.
4) In section 2.2, there should be additional description of the reaction conditions. Again, the reader should not be forced to go to the end of the manuscript to figure out what is going on. What are the concentrations of the reactants? H2:CO ratio? The boiling point of dioxane is 100 C. At 180 C, what fraction of this this is liquid and what is vapor? Do you need the solvent?
5) What is really missing from all of the reaction data is some benchmarking. How does one compare this to what is in the literature? Is there really anything special about the materials the authors have made? It is impossible to tell. There should have been some attempt to prepare a material that others in the literature have prepared. Without that, it is not clear what any of this means.
6) How different are conversions as a function of surface area? What happens when reaction conditions change? How does temperature affect selectivities? Do selectivities change with conversion (i.e. changing space time)? What is the effect of H2:CO ratio? What would happen if you didn’t have a solvent?
7) To me, the XPS data is completely uninformative. XPS is not particularly surface sensitive. Similarly, what is the point of the TGA analysis? Since such a low fraction of these materials are on the surface, most of the signal is coming from the bulk.
